# Disturbance in Mammalian Cognition Caused by Accumulation of Silver in Brain

**DOI:** 10.3390/toxics9020030

**Published:** 2021-02-03

**Authors:** Anna A. Antsiferova, Marina Yu. Kopaeva, Vyacheslav N. Kochkin, Pavel K. Kashkarov, Mikhail V. Kovalchuk

**Affiliations:** 1National Research Center “Kurchatov Institute”, 1, Akademika Kurchatova sq., 123182 Moscow, Russia; m.kopaeva@mail.ru (M.Y.K.); Kochkin_VN@nrcki.ru (V.N.K.); kashkarov_pk@nrcki.ru (P.K.K.); koval@nrcki.ru (M.V.K.); 2Moscow Institute of Physics and Technologies, 9, Institutskii Lane, Moscow Region, 141700 Dolgoprudny, Russia; 3Department of Physics, Lomonosov Moscow State University, 1, GSP-1, Leninskiye Gory, 119991 Moscow, Russia

**Keywords:** silver nanoparticles, contextual memory, accumulation in brain, biodistribution, hippocampus, cerebellum, cortex, neutron activation analysis, CA2, fear

## Abstract

The influence of daily prolonged administration of silver nanoparticles on the cognitive functions of a model mammal was studied. The accumulation of silver in the whole brain and the hippocampus, cerebellum, cortex and residual brain tissue of the mouse was investigated by highly precise and representative neutron activation analysis, and histological studies were conducted. Here, we show that long-term memory impairments were caused by the accumulation of silver nanoparticles in the brain and its subregions, such as the hippocampus, cerebellum and cortex, in a step-like manner by disturbance of hippocampal cell integrity. Three different approaches allowed us to observe this phenomenon and discover the reasons it occurred.

## 1. Introduction

Since the beginning of the 2000s, silver nanoparticles (Ag-NPs) have been used in various applications from pharmaceutics [1,2,3,4] to textiles [5,6,7] and from hygiene products [8] and packing materials [9] to daily food supplements [10,11,12,13,14]. Undoubtedly, Ag possesses unique antiseptic properties that have been known since antiquity [15]. For example, people in the past successfully used silver dishes as antiseptics [16].

Currently, silver has not been forgotten and still competes with modern antibiotics [17,18].

Why use Ag-NPs rather than bulky silver? Nanosized particles demonstrate similarities to natural compounds and their corresponding effects. During evolution, mechanisms developed to address natural nanosized structures such as proteins and cellular organelles. For instance, a living cell can recognize proteins in the native state [19], which are always nanosized, as solid or liquid food and take them up. Therefore, engineered nanoparticles with or without a corona of proteins can be regarded as food or construction materials for the cell. The manner of uptake, such as phagocytosis [20] and micropinocytosis [21], of any nanosized structure has been developed by nature. This results in the high cellular penetrability of engineered nanoparticles, their ability to accumulate inside cells and their bioactivity.

For instance, in some cases, Ag-NPs can accumulate in different organs, such as the liver, kidney, blood and brain of mammals [22,23,24,25]. However, the route of exposure plays a major role in the targeting of different organs [26]. Several works have shown that all tissues except the brain easily remove silver [23,24], and some scientific studies have reliably demonstrated the penetration of Ag-NPs via the blood-brain barrier and their uptake by nerve tissue [23,24,27,28]. Therefore, the brain can be regarded as a target organ for Ag-NPs.

Many scientific works have shown the toxicity of Ag-NPs to nerve cells, such as neurons, astrocytes and glial cells [29,30,31,32,33,34,35]. Perhaps, the toxic effect is directly proportional to the administered dose of Ag-NPs and the exposure time as it has been previously shown in many studies [36,37].

Due to the accumulation of Ag-NPs in the brain and their toxicity to brain cells, some recent works have been devoted to studying the influence of Ag-NPs on the cognitive and behavioral functions of model mammals such as rats and mice [38,39,40,41,42,43]. Usually, such studies are conducted for no more than 30 days, and the doses of Ag-NPs are able to poison animals without acting on the brain [44].

The objective of the present research was to comprehensively study the influence of low concentrations of polyvinylpyrrolidone-coated (PVP-coated) nanoparticles on mammalian cognitive functions, quantitatively describe the biokinetics of Ag-NPs in the brain, determine the amounts of silver in different parts of the brain, such as the cerebellum, hippocampus, and cortex, and perform morphological studies of brain slices. All the studies were conducted under prolonged 1- to 6-month oral administration of Ag-NPs, implemented with the use of neutron activation analysis (NAA).

## 2. Materials and Methods

### 2.1. Nanoparticles

A colloidal solution of Ag-NPs was provided by the food supplement Argovit C manufactured in the Russian Federation, Novosibirsk and was used as a nanosilver. It was packed in an opaque, plastic 0.5 l bottle. The nanoparticles were coated with PVP, and the initial concentration of silver in the solution was 10 µg/mL in distilled water.

Dynamical light scattering (DLS) (Malvern Zetasizer Nano ZS) and transmission electron microscopy (TEM) (Titan 80–300, Thermo Fisher Scientific) were used to measure the size and stability of the nanoparticles. TEM was also applied to visualize the nanoparticles. For this purpose, the initial solution was dissolved in distilled water to a concentration of 0.1 µg/mL for a total volume of 15 mL. Half of the solution was used for immediate measurements, and the remaining part was preserved in the refrigerator at +2 °C for 1 year and then investigated with DLS and TEM to assess the stability of the Ag-NPs.

For the immediate measurements, the solution was divided into 2 equal parts, one for DLS and the other for TEM. Before performing the measurements, the solutions were sonicated in an ultrasonic bath for 15 min. For DLS, 1 mL of the solution per measurement was poured into a plastic cuvette, and a series of measurements was made. For TEM, 0.01 mL of the solution was applied to a carbon grid, dried and measured microscopically. The same experiment was performed with the preserved solution.

### 2.2. Nanoparticle Exposure Mosel

Eighty-eight C57BL/6 eight-week-old male mice weighing 19–27 g were purchased from a supplier in Stolbovaya, Moscow, Russia. Mice were kept in individual ventilated cages with access to standard laboratory food and water ad libitum with room temperature controlled at 23 ± 2 °C and a 12/12 h light/dark cycle. All experimental procedures were performed in accordance with the rules of the Ministry of Health of the Russian Federation (№ 267 of 19.06.2013) and the Local Ethics Committee for Biomedical Research of the National Research Center “Kurchatov Institute”. Scheme 1 presents the general route of the experiment with the laboratory mammals.

Animals were randomly divided into four experimental groups, 20 mice per group: ‘30 days’, ‘60 days’, ‘120 days’ and 28 for ‘180 days’. Each day a half of the amount, 10 (14) animals in each group were orally administered Ag-NPs suspended in distilled water (50 µg per day) throughout the whole experiment. The mice received the solution in their drinking water. The dose was chosen as ½ of the previously used dose for NAA studies [19,20]. This allows tracing of the accumulated amount of NPs in the different regions of the brain and ensures the mice do not die. The corresponding amount of Ag-NPs could be consumed by a human as a daily food supplement or through tap water as long as the water was not recycled or filtered to remove NPs. It is difficult to estimate the amount of nanoparticles in the water supply system. The rest of the animals received sterile water.

Body weight was monitored weekly during the whole exposure period.

### 2.3. Cognitive Test

To examine the effect of silver nanoparticles on memory formation and retention, the mice were trained and their memory was tested in a contextual fear conditioning task using a Video Fear Conditioning System (MED Associates Inc., St. Albans, VT, USA) and Video Freeze software v2.5.5.0 (MED Associates Inc., St. Albans, VT, USA). Video recordings of the animals’ behavior were made during training and testing. The number and duration of freezing acts were determined automatically.

Animals from each experimental group (Ag-NP-exposed mice and control mice separately) were randomly assigned to two subgroups: ‘fear conditioning’ (FC) (*n* = 6) and ‘active control’ (AC) (*n* = 4) (‘30 days’, ‘60 days’, ‘120 days’) and (FC) (*n* = 8) and (AC) (*n* = 6) (‘180 days’). Mice from the FC subgroup were placed for 6 min into the experimental chamber, where they freely explored the new environment for 3 min, and then 3 electric foot shocks (1 mA, 2 s) were delivered with a 1-min interval followed by a 1-min rest period. Mice from the AC subgroup were placed for 6 min into the experimental chamber, where they freely explored the new environment without a foot shock. The animals learned just once at the end of the exposure period. Mice were returned to their home cages immediately after the training. Twenty-four hours after training, the animals were tested for long-term memory retention (3 min in the experimental chamber without a foot shock). Recollection of the chamber environment induced an association between the chamber and the negative feeling of foot shocks in the FC group, while the animals in the AC group made no such connection. The proportion of freezing acts versus test duration was assessed as a measure of long-term memory. Before placing each animal into the chamber, the chamber was wiped with 70% ethanol.

Animals were compared in terms of the following parameters: percentage of freezing acts before and after the current was applied during training and the percentage of freezing acts during testing.

Statistical analysis for the cognitive test was performed with GraphPad Prism 6 by the nonparametric Mann-Whitney test. The differences were considered significant at *p* < 0.05. All data are expressed as the median ± interquartile range.

### 2.4. Tissue Analysis

After contextual memory testing of each experimental group, animals were divided into 2 groups: biokinetic studies (BS) (8(12) experimental and 8(12) control mice in each subgroup) and histopathological studies (HS) (2 experimental and 2 control mice in each subgroup). The mice in the BS group were divided into 2 subgroups: ‘whole brain biokinetics’ (WBB) (4(6) experimental mice and 4(6) control mice) and ‘brain parts biokinetics’ (BPB) (4(6) experimental mice and 4(6) control mice).

Mice were deeply anaesthetized and sacrificed by decapitation. The brains were rapidly removed from the skulls, postfixed for 24 h in 1% paraformaldehyde, and dissected. The hippocampus, cortex, cerebellum and remaining brain tissue were isolated. Then, WBB and BPB samples were placed into marked individual opened plastic tubes and dried at 75 °C for 48 h until desiccated.

HS mice were anaesthetized and perfused with 4% paraformaldehyde in phosphate buffered saline (pH 7.4). The brains were then removed and postfixed. Forty micrometer coronal sections were cut through the whole brain on a Leica VT1200S vibratome. For the HS brains, every sixth section was Nissl-stained with cresyl violet and examined by light microscopy (Zeiss Imager z2 VivaTome, Carl Zeiss, Germany).

Samples in marked closed plastic tubes (Eppendorf AG, Hamburg, Germany) were put layer by layer in aluminum cases with the reference samples whose silver content was known, as shown in Figure 1. The reference samples were prepared just prior to use. For this purpose, a standard sample of silver with a known mass was poured onto sterile cotton wool, and then reference samples with the same geometry as the experimental samples were prepared and placed into plastic tubes. Additionally, known amounts of state standard samples were poured onto paper rings, dried and placed between the layers. Thus, one aluminum case contained one reference sample with the same geometry as the experimental samples and 84 ‘flat’ reference samples. The cases were put into vertical channels of an IR 8 nuclear research reactor (power 8 MeW) and irradiated for 24 h at a neutron flux of 10^−12^ neutron/sec × cm^2^ to obtain the necessary activity of the ^110m^Ag radioisotope during NAA.

Then, all the WBB and BPB experimental and reference samples were examined by gamma-spectrometric elemental analysis (CANBERRA) to measure the activity of silver in the samples. The amount of silver in the experimental samples was calculated by comparing their silver activity with that of the reference samples. Methodological errors were ignored because of their low value. All data are expressed as the means ± SEM. Statistical analysis test was performed by the nonparametric Mann-Whitney test. The differences were considered significant at *p* < 0.05.

## 3. Results

### 3.1. Nanoparticles

Dynamic light scattering and transmission electron microscopy demonstrated that the size of the nanoparticles was 34 ± 5 nm and that they had quasi-spherical shapes (Figure 2). As described in reference [10], such PVP-coated nanoparticles are very stable (no significant size changes during one year of refrigeration) and can pass through the blood-brain barrier [22], which is due to the high stability of the Ag-NPs, their size and, perhaps, their hydrophobic coating. This hydrophobic coating may provide an attachment to the cellular membranes and better penetrability into some types of cells.

### 3.2. Accumulation of Silver Nanoparticles in Brains and Their Subregions

The amount of silver that had accumulated in the brains and their subregions was studied by NAA.

NAA demonstrated that the amount of silver in the control samples was below the critical recognition threshold, so they were set to zero. However, the amount of silver in the brains and their subregions of the experimental animals were determined with an error of no more than 10%.

The accumulation of silver in the mouse brains during the period of administration could be approximated as a monotonously increasing function that approached a constant value (Figure 3). The function indicates that silver gradually accumulated in the mouse brain following daily constant administration of Ag-NPs. This result is similar to the data from our previous research [21].

Figure 4 shows the histograms of the silver concentration in the subregions of the brain as a function of the period of administration. It can be observed that the concentration increases at specific times in a step-like manner. The steps increases for the hippocampus (a) (*p* = 0.032), cerebellum (b) (*p* = 0.046), and cortex (only tendency is observed) can be seen at 120 days and for the remaining brain tissue at 180 days (*p* = 0.048).

Figure 3 does not show any step-like increases because it plots the concentration for the whole brain, which consists of all the subregions described above; consequently, the two step-like increases are combined, resulting in a function with more gradual increases.

### 3.3. Morphological Studies

The necropsy observations indicated that compared to those of the control mice, all the brains of the Ag-NP-exposed mice exhibited the expected anatomic features (e.g., color, consistency, shape, and size).

Analyses of the data obtained from measuring the body weights and brain weights of mice revealed that Ag-NPs had no significant effect on either measurement (Table 1).

According to the Nissl staining, no apparent structural differences were found in the whole brain, particularly in the hippocampus, between Ag-NP-exposed and control mice in the ‘30 days’ and ‘60 days’ groups. We examined the amygdala region of the brain but found no differences in neuronal density for any of the experimental groups.

However, visible changes were seen in the CA2 region of the hippocampus of Ag-NP-exposed mice with respect to that of the control mice in the ‘120 days’ and ‘180 days’ groups. The stratum pyramidale showed an irregular and rarefied appearance, and the CA2 region in general presented a dispersed arrangement of neurons after exposure to Ag-NPs (Figure 5).

No changes were found for cerebellum and cortex for all the periods.

### 3.4. Influence of Nanoparticles on Cognitive Functions

The influence of Ag-NPs on behavioral and cognitive functions was described in our previous research [10]. It was shown that affected mammals go through three different periods: anxiety, activation of the mechanism of adaptation and disturbance of cognitive functions. The initial slight disturbance in behavior, which was thoroughly described in [10], may be associated with the accumulation of silver in the brains and the development of oxidative stress and could be one of the possible mechanisms through which Ag-NPs influence cells, as authors have indicated in different living systems [33,45,46,47]. However, these initial changes do not seem to be fatal during this period because of self-adaptation. This adaptation mechanism does counteract the negative effect of Ag-NPs over longer durations, as seen below.

Regarding long-term contextual memory, no changes in learning were detected for any period.

Figure 6 demonstrates the effect of 30-, 60-, 120- and 180-day exposure to Ag-NPs on contextual memory, particularly the number of freezing acts during memory testing of FC groups. The memory was reliably formed in each case [10]. Figure 6 shows that there is no reliable difference in memory between the control groups and Ag-NP groups at 30 (Mann-Whitney test: *U* = 15.00, *p* = 0.675), 60 (*U* = 12.00, *p* = 0.387) and 120 (*U* = 10.00, *p* = 0.238) days. A reliable difference can be seen at 180 days of administration, when Ag-NP mice made a statistically smaller number of freezing acts than control mice (*U* = 8.000, *p* = 0.043).

This is indicative of the degradation of long-term contextual memory that formed as a result of exposure to the chamber and secured by the very strong conditioning from the electric pulses delivered to the feet. The mice apparently did not remember the negative feelings they were exposed to the day before during testing.

## 4. Discussion

For the first time we applied a complex approach and long-term periods of Ag-NPs exposure that has not been attempted by other authors before.

The use of the three different approaches allowed us to determine how Ag-NPs influence the cognitive functions of mammals and to discover the possible reasons for the trigger of the phenomena such as the accumulation of silver in the whole brain and its subregions and thus induced disturbance of the CA2 region.

NAA is undoubtedly a highly promising tool for nanoparticle biokinetics observation that provides an opportunity to precisely measure the amounts of chemicals accumulated in organic and inorganic media. In comparison to electron microscopy and mass spectrometry [43], it is a very representative method because it allows us to work with whole organs and their subregions, obviating the needs for microsection required in more widely used techniques. This is one of the reasons for its sensitivity because it is easy to obtain gamma- or beta- signals from whole samples such as organs. More precisely, however, the sensitivity depends on the correct choice of the irradiation time, which is necessary to achieve the proper activity, and, of course, on the properties of the radioisotope, such as the half-life and cross-section. For instance, the half-life of the ^110m^Ag radioisotope is approximately 250 days, which allows us to collect information about the activity of silver within this period and even longer if the activity is not very high.

It should be noted that the NAA data obtained in our research describing the amount of silver accumulated in different subregions of the brain are unique; interestingly, two-fold and even three-fold step increases were detected. Previously obtained results on the excretion of silver from the brain demonstrated that the elimination rate increased after 120 days of Ag-NP administration and predicted early changes in cell integrity, in complete correspondence to the present data [21]. The accumulation of silver in the brain subregions and the observed step increases may be due to the loss of cell integrity and the increase in excretion speed. However, an increase in excretion speed may also be regarded as the loss of silver from the brain tissue. In this case, the uptake should also be higher, which was proven by the growth of the mass of silver in the whole brain and its subregions. The low-density organization of neurons in the CA2 subregion observed at 120 and 180 days proves the loss of cell integrity.

The CA2 region is responsible for informational memory and recognition, allowing the organism to distinguish certain environments, situations and circumstances and to make connections between them. Recent findings have shown that CA2 is involved in novelty detection [48], social memory [49], and ripple generation [50]. CA2 has widespread anatomical connectivity, unique signaling molecules, and intrinsic electrophysiological properties. CA2 transmission plays a role in spatial learning, perhaps through its influence on overall network excitability [51].

Damage to CA2 leads to network disturbance, difficulties with connection to the environment and the situation, degradation of memory and other unpredictable effects. It was shown in the present research that the damage to CA2 was accompanied by reliable long-term contextual memory degradation [52]. Additionally, it is well known that the hippocampus is responsible for neurogenesis; therefore, damage to this region may have a negative effect on stem cells, slowing neurogenesis [53]. The results were obtained with a very ‘strong’ memory test, which is based on creating fear of an environment by causing pain. Detection of memory changes in this test are definitive evidence of cognitive damage. However, it should be noted that changes to the CA2 subregion together with the step-like increase in silver content in the hippocampus were first detected two months before the memory impairment occurred.

It is clear that the accumulation of silver in the brain and in the hippocampus specifically as well as the low-density organization of neurons in the CA2 subregion led to the memory disturbances.

It is interesting that no morphological changes have been observed in other brain regions. Taking into account the fact that silver is accumulated and distributed in the brain unevenly preferring certain regions we may speculate that Ag-NPs are selective for the regions. It can be due to architectural and anatomic features of the tissues as well as their functions caused by metabolic processes type. Perhaps, a specific threshold of Ag-NP concentration should take place determining morphological and future behavioral (cognitive) changes.

Therefore, it is obvious that the daily consumption of low doses of Ag-NPs may be dangerous from different points of view because of possible memory and behavioral changes, which are caused by the accumulation of silver in the brain and its subregions and to integrity changes in the CA2 subregion of the hippocampus. Use of Ag-NPs as a food supplement and a drug for humans should be careful. It is clear now that Ag-NP uptake should not be too durable. Specific times may be determined after quantitative transfer of the data to humans made after development and application of a valid model.

## Data Availability

The work fulfills the Ethical Guidelines for Authors.

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
