# Peer review of "Disturbance in Mammalian Cognition Caused by Accumulation of Silver in Brain"

_toxics, 2021, doi:10.3390/toxics9020030_

Round 1
Reviewer 1 Report
Antsiferova et al. present important and interesting findings regrading the impact of silver accumulation in brain parenchyma, its impact on neuronal tissue and on cognitive function.
My major concern addresses the cognitive test performed, which suggests long-term contextual, fear-related, memory loss, and its possible relation to Ag-NP accumulation in the brain. My comments are as follows:
- The conclusions are based on a relatively low sample size (n=6) for behavioural studies. Additionally, results of the memory test (Figure 2) are similar in pattern to the accumulation of Ag-NP’s in the hippocampus (Figure 4a), which does not indicate gradual, dose-dependent accumulation, but rather low and high accumulations in response to short (30d and 60d) and long (120d and 180d) exposure respectively. Therefore, my suggestion is to either increase the sample size of the memory test, or alternatively, unify findings from 30d with those from 60d (short exposure), and from 120d with those from180d (long exposure). Similarly, Ag-NP accumulation values in the hippocampus should be unified as well.
- Please add statistical analysis results (significance values, which test was used) to the text related to figures 2, 3 and 4.
- In Figure 2 the legend indicates that both Ag-NP and control (no exposure to Ag-NP) groups were fear-conditioned. However, the related text implies that the control group was not fear-conditioned: " The contexts were the combination of electric pulses and the chamber for the Ag-NP groups or the chamber alone for the control groups" (lines 183-184). Please clarify.
- Why does the legend for Figure 2 indicate the sample size as n=6-8 per group? According to the Methods section, the sample size should be n=6. Please clarify.
- In line 281-284, the statement made is unfounded and no citation is given to support it. The directionality of molecular traffic across the BBB is more complex than suggested in this statement, and the sliver accumulation-related neuronal damage, evident in CA2, suggests only that and nothing more. Please add citations for support or rephrase.
Some minor comments:
- Lines 256-258: The authors’ findings describe the phenomena, not the reasons or underlying mechanisms for its occurrence.
- Lines 295-297: "Additionally, it is well known that the hippocampus is responsible for neurogenesis; therefore, damage to this region may have a negative effect on stem cells, slowing neurogenesis". Please provide citations to support this statement.
- In Figure 5, please add numerical scale to images.
- Lines 187-189: "It seems that the adaptation mechanism was activated during this period of Ag-NP administration because the memory of the Ag-NP mice seems to have improved. Additionally, the results may indicate that the mice are somehow ‘acute’ during this time". This statement is more suited for the Discussion section than for the Results.
- Lines 165-167: "…which is due to the high stability of the Ag-NPs, their size and, perhaps, their hydrophobic coating. This hydrophobic coating may provide an attachment to the cellular membranes and better penetrability into some types of cells". This statement is more suited for the Discussion section than for the Results.
- Lines 169-180: This paragraph is more suited for the Discussion section than for the Results.
- In Figure 3, it seems that the curve indicates saturation for long exposure. In my opinion, this could be addressed in the Discussion.
- Lines 135-139: This paragraph addresses the BS group and therefore should be placed right after the mentioning of this group and prior to the mentioning of the HS group.
- Line 44: The word "jamming" could possibly be replaced by "uptake", to better describe the biological phenomena.
- If possible, please add an image of the nanoparticles, acquired by TEM.
- A more logical order of presentation of findings should be, in my opinion: biokinetic evaluation, followed by histological evaluation, and finally cognitive tests.
Author Response
Thank you for the detailed perusal and a high rating of our work. We took into the account all your comments and tried to fulfill them. It improved quality of the manuscript.
Let us give responses to your comments step by step:
Major comments
- We used a non-parametric Mann-Whitney test to compare values in the case, which allows applying data rows with a relatively low number of measured values. We emphasized the fact that the accumulation is step like rather than gradual. ‘Figure 4 shows the histograms of the silver concentration in the subregions of the brain as a function of the period of administration. It can be observed that the concentration increases at specific times in a step-like manner’. However, the kinetics of silver accumulation in the whole brain is gradual which we described in the text. ‘Figure 3 does not show any step-like increases because it plots the concentration for the whole brain, which consists of all the subregions described above; consequently, the two step-like increases are combined, resulting in a function with more gradual increases’. Moreover, we observe step increases for hippocampus and cerebellum since 120 days and for cortex and the remaining part of brain since 180 days. Thus, unification of the data would not be quite rightful as it could break the whole ‘picture’.
- We added significance values to the text related to figures where p<0.05. We indicated that ‘Statistical analysis test was performed by the nonparametric Mann-Whitney test. The differences were considered significant at p < 0.05'. Fig. 3 demonstrates gradual increase.
- We are sorry for this misunderstanding in the ‘control’ groups. A part of Ag-NP groups and a part of control groups were exposed to electric pulses. The rest of the animals was not exposed to electric pulses as it is indicated in Material and Methods. We changed this sentence to ‘The contexts were the combination of electric pulses and the chamber for the FC groups or the chamber alone for the AC groups and added some clarifications to the text’.
- We forgot to indicate the fact and did not indicate the fact that ‘180 days’ group included 28 animals not just 20 as it is valid for all other groups. We added it to Materials and Methods. Thus, we had initially 88 mice, not just 80. We made all the required changes to Materials in Methods where it’s needed in brackets.
- It is our supposition that comes from an obvious speculative model. We added the clarification ‘Due to a simple speculative model if all the gates are open, both streams should circulate in and out’ to the text.
Minor comments
- We rephrased the sentence to “The use of the three different approaches allowed us to determine how Ag-NPs influence the cognitive functions of mammals and to discover the possible reasons for the trigger of the phenomena such as the accumulation of silver in the whole brain and its subregions and thus induced disturbance of the CA2 region’. Our supposition is that accumulation of silver causes CA2 damage and cognitive function repairment what we tried to describe in the manuscript.
- We supported the statement by Das, S., Basu A. Inflammation: A New Candidate in Modulating Adult Neurogenesis. Journal of Neuroscience Research. 86.1199–1208 (2008). doi:10.1002/jnr.21585.
- We added numerical scales to images. The scales for A, C, E are the same. The scales for B, D, F are the same.
- We agree with the statement. However, the transfer would not change the sense of the paper.
- We agree with the statement. However, the transfer would not change the sense of the paper.
- We agree with the statement. However, the transfer would not change the sense of the paper.
- We agree with the statement. However, the transfer would not change the sense of the paper.
- The mentioned paragraph was placed prior to HS group description.
- We rephrased the sentence to ‘Several works have shown that all tissues except the brain easily remove silver [23,24], and some scientific studies have reliably demonstrated the penetration of Ag-NPs via the blood-brain barrier and their uptake by nerve tissue [23, 24, 27, 28]’.
- We added the TEM image to the manuscript.
- We changed the order in Results according to the suggestion. The order in Materials and Methods cannot de changed due to mice could not be changed due to mice behavior cannot be tested after scarification. The story is told gradually.

Reviewer 2 Report
This is a nicely written study of the influence of prolonged administration of silver nanoparticles on the cognitive functions of mice. The accumulation of silver in the whole brain, hippocampus, cerebellum, cortex, and residual brain was investigated by the method of neutron activation analysis, which was accompanied by histological and memory studies.
The paper is nice to read and experiments are described in detail.
I have two issues:
- I am not clear about the dose you administered. In l.90 you state 1/2 of the previous dose. In l. 210 pp. you write constant dose. In Fig. 2 caption, add daily doses.
- The discussion should be extended and sharped: The combination of three methods (especially with NAA) is new. Also the separated study of different brain regions. I find no comment on morphological studies of cerebellum and cortex. If there are no obvious changes, discuss the possible reasons. Is there a threshold for the effective concentration? Are NP-Ag selective? Why do different brain regions show different Ag concentrations? Please try also to include results from literature. Discuss the transferability of the results to humans.
minor
l.29 - l.38 please rephrase this section;
f.e: why use AG NP rather than silver alone ?-> why use AG NP rather than bulky silver.
l. 41 space missing
References - the format is not consistent f.e. (year)
Author Response
Thank you for the detailed perusal and a high rating of our work. We took into the account all your comments and tried to fulfill them. It improved quality of the manuscript.
Let us give responses to your comments step by step:
Major comments
- Sorry for the wrong sentence. We rephrased it to ‘The mice received the solution in their drinking water. The dose was chosen as ½ of the previously used dose for NAA studies [19, 20]’. Thus, as it is said above ‘Each day a half of the amount, 10(14) animals in each group were orally administered Ag-NPs suspended in distilled water (50 µg per day) throughout the whole experiment’.
- The combination of three methods (especially with NAA) is new. Also the separated study of different brain regions. I find no comment on morphological studies of cerebellum and cortex. If there are no obvious changes, discuss the possible reasons. Is there a threshold for the effective concentration? Are NP-Ag selective? Why do different brain regions show different Ag concentrations? Please try also to include results from literature. Discuss the transferability of the results to humans.
We widened the discussion:
‘It is interesting that no morphological changes have been observed in other brain regions. Taking into account the fact that silver is accumulated and distributed in the brain unevenly preferring certain regions we may speculate that Ag-NPs are selective for the regions. It can be due to architectural and anatomic features of the tissues as well as their functions caused by metabolic processes type. Perhaps, a specific threshold of Ag-NP concentration should take place determining morphological and future behavioral (cognitive) changes’.
‘Use of Ag-NPs as a food supplement and a drug for humans should be careful. It is clear now that Ag-NP uptake should not be too durable. Specific times may be determined after qualitative transfer of the data to humans made after development and application of a valid model.’
- 29 - l.38 please rephrase this section;
f.e: why use AG NP rather than silver alone ?-> why use AG NP rather than bulky silver.
We made the suggested amendment.
- 41 space missing
We added a space.
- References - the format is not consistent f.e. (year)
We changed the format in References (year).

Reviewer 3 Report
The paper describes the disturbance in mammalian cognition caused by accumulation of silver in brain. The authors observe the influence of daily prolonged administration of silver nanoparticles on the cognitive functions of a model mammal. The results are scientifically valid and the manuscript is well-written, but the authors need to improve methodology and justify relevance of their studies in comparison with other previous reports.
Line 41: Please make a space between “shown that”.
Lines 47-48: Please add a reference for this statement.
Lines 61-158: The studies carried out are very complicated. While reading the text, I made a diagram that helped me understand the number and names of the groups, as well as the number of animals per group. Please add a figure (diagram, draft or schema) to the section Material and Methods, that shows the model of the experiment.
Please, divide the chapter into subsections (2.1., 2.2. etc.) that will relate to the subsections of the section 3.Results.
Lines 234-248: Please add a title and description for the Table 1.
Lines 293-295: Please add a reference for this statement.
Lines 295-297: Please add a reference for this statement.
Author Response
Thank you for the detailed perusal and a high rating of our work. We took into the account all your comments and tried to fulfill them. It improved quality of the manuscript.
Let us give responses to your comments step by step:
- Line 41: Please make a space between “shown that”.
A space is added there.
- We slightly rephrased the statement and added references.
Perhaps, the toxic effect is directly proportional to the administered dose of Ag-NPs and the exposure time as it has been previously shown in many researches [36, 37].
Park, E.-J., Bae, E., Yi, J., et al, Repeated-dose toxicity and inflammatory responses in mice by oral administration of silver nanoparticles. Environmental Toxicology and Pharmacology, 2010, 30, 162-168. doi:10.1016/j.etap.2010.05.004.
Garcı´a-Contreras, R., Argueta-Figueroa, L., Mejı´a-Rubalcava, C., et al, Perspectives for the use of silver nanoparticles in dental practice. International Dental Journal, 2011, 61, 297–301. doi: 10.1111/j.1875-595X.2011.00072.x.
- Lines 61-158: The studies carried out are very complicated. While reading the text, I made a diagram that helped me understand the number and names of the groups, as well as the number of animals per group. Please add a figure (diagram, draft or schema) to the section Material and Methods, that shows the model of the experiment.
We added Scheme 1 showing the general route of the experiment with mammals.
Please, see the attachment.
- Please, divide the chapter into subsections (2.1., 2.2. etc.) that will relate to the subsections of the section 3.Results.
We divided the chapter into suitable subsections
- Lines 234-248: Please add a title and description for the Table 1.
We added a title for the Table 1. The description has been already presented in the previous version and still remains. ‘Analyses of the data obtained from measuring the body weights and brain weights of mice revealed that Ag-NPs had no significant effect on either measurement (Table 1). ‘
- Lines 293-295: Please add a reference for this statement.
The reference is added.
Chi-Ching Pang, C., Kiecker, C., O’Brien, J. T., et al, Ammon’s Horn 2 (CA2) of the Hippocampus: A Long-Known Region with a New Potential Role in Neurodegeneration. The Neuroscientist, 2018, doi:10.1177/1073858418778747
- Lines 295-297: Please add a reference for this statement.
The reference is added.
Das, S., Basu A. Inflammation: A New Candidate in Modulating Adult Neurogenesis. Journal of Neuroscience Research, 2008, 86. 1199–1208. doi:10.1002/jnr.21585.
Also we added the phrase to Discussion to justify the relevance of the research. ‘For the first time we applied a complex approach and long-term periods of Ag-NPs exposure that has not been attempted by other authors before’.

Round 2
Reviewer 1 Report
All of my minor comments were addressed in full.
With regards to my major comments:
- Using non-parametric statistic evaluations does not validate working with a small sample size. The latter requires the former. Even though statistical significance is observed, it does not undoubtedly confirm a physiologic/cognitive impact. Especially, if the result is not extremely significant, which is the case for the current finding (p=0.043). It is well-agreed that behavioural tests require N>10, in order to make proper conclusions. However, it is possible to present data based on low-sample size, provided the size is mentioned in the text, which is clearly the case for this manuscript. My suggestion to unify groups was made based on the numerical pattern observed, and for the observations to be based on a larger sample size, per-group. It is of course, not obligatory in terms of scientific soundness.
- This comment was addressed in full.
- If I understand correctly, the statement regarding AC groups in lines 259-260 does not refer to anything displayed in figure 6 and is therefore redundant. I suggest removing it, as it may confuse the reader.
- This comment was addressed in full.
- The issue of directionality of molecular traffic across the BBB is far too complex to make conclusions based on simple speculations, that are unsupported in the relevant literature. I suggest removing this statement.
Author Response
Thank you. We appreciate your valuable comments that allowed improving our work.
1. Using non-parametric statistic evaluations does not validate working with a small sample size. The latter requires the former. Even though statistical significance is observed, it does not undoubtedly confirm a physiologic/cognitive impact. Especially, if the result is not extremely significant, which is the case for the current finding (p=0.043). It is well-agreed that behavioural tests require N>10, in order to make proper conclusions. However, it is possible to present data based on low-sample size, provided the size is mentioned in the text, which is clearly the case for this manuscript. My suggestion to unify groups was made based on the numerical pattern observed, and for the observations to be based on a larger sample size, per-group. It is of course, not obligatory in terms of scientific soundness.
We agree with the comment. Yes, the text includes all the data on the sample sizes. We had 6 FC and 6 AC animals in each (Ag-NP and control) group for ’30 days’, ’60 days’ and ‘120 days’. In addition, we had 8 AC and 8 FC animals in each (Ag-NP and control) group for ‘180 days’. We indicated additional statistical data in the related text. ‘Figure 6 shows that there is no reliable difference in memory between the control groups and Ag-NP groups at 30 (Mann–Whitney test: U = 15.00, p = 0.675), 60 (U = 12.00, p = 0.387) and 120 (U = 10.00, p = 0.238) days. A reliable difference can be seen at 180 days of administration, when Ag-NP mice made a statistically smaller number of freezing acts than control mice (U = 8.000, p=0.043)’.
Also we decided to remove the text: ‘However, at 120 days, there was a tendency for Ag-NP mice to perform more freezing acts than control mice. It seems that the adaptation mechanism was activated during this period of Ag-NP administration because the memory of the Ag-NP mice seems to have improved. Additionally, the results may indicate that the mice are somehow ‘acute’ during this time.’
2. If I understand correctly, the statement regarding AC groups in lines 259-260 does not refer to anything displayed in figure 6 and is therefore redundant. I suggest removing it, as it may confuse the reader.
We have removed the confusing statement.
3. The issue of directionality of molecular traffic across the BBB is far too complex to make conclusions based on simple speculations, that are unsupported in the relevant literature. I suggest removing this statement.
We removed the statement and rephrased the last sentence. From ‘The low-density organization of neurons in the CA2 subregion observed at 120 and 180 days proves the prediction of opening the gates.’ To ‘The low-density organization of neurons in the CA2 subregion observed at 120 and 180 days proves the loss of cell integrity’.
